# Application of Chitosan and Its Derivatives in Postharvest Coating Preservation of Fruits

**DOI:** 10.3390/foods14081318

**Published:** 2025-04-11

**Authors:** Limin Dai, Xiaoshuai Wang, Jun Zhang, Changwei Li

**Affiliations:** 1School of Agricultural Engineering, Jiangsu University, Zhenjiang 212013, China; lmdai@ujs.edu.cn; 2College of Biosystems Engineering and Food Science, Zhejiang University, Hangzhou 310058, China; xiaoshuai.wang@zju.edu.cn; 3School of Mechanical and Electrical Engineering, Jiaxing Nanhu University, Jiaxing 314001, China; 11613006@zju.edu.cn

**Keywords:** nanocomposite, modification, surface coating, active groups

## Abstract

Postharvest preservation of fruits is one of the key issues in the current agriculture and food processing industry. Surface coating treatment, a promising technology for postharvest fruit preservation, has gathered significant attention due to its ability to reduce water loss, regulate gas exchange, and inhibit respiration, thereby achieving postharvest fruit preservation. Among them, chitosan-based coating has a wide application prospect due to its superior film-forming capability, high biosecurity, wide range of sources, etc. This review summarizes the structural features, physicochemical properties, modification strategies, and preservation mechanisms of chitosan-based coatings, focusing on their applications in postharvest fruit storage. Unlike prior works, it highlights advanced modifications (e.g., nanocomposite, multifunctional grafting) that enhance antimicrobial activity, mechanical strength, and environmental adaptability. Challenges in fruit preservation—such as microbial resistance and stability—are analyzed, with solutions proposed via material innovation. The discussion on industrial scalability emphasizes chitosan’s biodegradability, cost-effectiveness, and alignment with sustainable agriculture, while addressing technical bottlenecks. This work bridges fundamental research and practical use, advancing chitosan-based coatings toward greener, safer, and scalable postharvest solutions.

## 1. Introduction

Fruit cultivation and production play an important role in agricultural production around the world [1]. However, postharvest fruits are susceptible to water loss, shrinkage, softening, and mildew due to the influence of their own physiological and metabolic activities and microbial infection [2,3,4,5], resulting in a large number of losses, especially in developing countries [6]. Therefore, to ensure a green and healthy supply of fruits, it is necessary to develop effective preservation techniques to improve fruit quality during storage [7,8,9,10]. At present, the common fruit preservation technologies are mainly divided into three categories: physical, biological, and chemical methods [11].

Physical preservation technology mainly includes low temperature [12,13], modified atmosphere [14,15], sonication [16,17], and irradiation [18,19] treatment methods, which have the advantages of high safety and substantial preservation effect [20,21]. However, physical preservation technology usually requires professional large-scale equipment, and the investment cost is pretty high [22], making it difficult to popularize it on a large scale. Biological preservation technology mainly includes microbial antagonists [23,24], natural plant extracts [25,26], and essential oil [27,28] treatment methods, which has the advantages of good biosafety and broad-spectrum antimicrobial activity [29,30]. However, biological preservation technology is susceptible to external environmental interference, and the preservation effect is difficult to maintain for a long time [31]. Chemical preservation technology mainly includes calcium [32], 1-methylcyclopropylene [33], and edible coating [34], which has the advantages of good preservation effect [35,36,37]. However, the disadvantages of chemical preservation technology are that there are residues in use, and the safety needs to be carefully evaluated [38].

Over the past few decades, edible coating technology has developed rapidly in the food industry [39,40]. Edible coating materials are widely used, and their sources are mainly polysaccharides [41], proteins [42], and lipids [43]. By spraying [44], brushing [45], or soaking [46], the edible coating material adheres directly to the surface of the fresh produce, and the fresh produce receives an additional protective coating, which effectively maintains the quality during storage [47,48,49]. Therefore, chitosan is widely used in meat [50], cheese [51], and fruit [52] preservation. In addition, the added active substances can give the coating desired mechanical properties, antimicrobial properties, and antioxidant properties [53,54]. Due to the good film-forming capability, biocompatibility, biological activity, and broad-spectrum antimicrobial effect of chitosan, it can reduce the infection rate of fruit and is often used as a matrix for the surface coating of fruits [55,56]. Compared to refrigeration, chitosan coatings eliminate the need for extensive cold chain infrastructure and provide sustained preservative efficacy after application. In contrast to traditional chemical methods, they offer distinct advantages: abundant natural sources, excellent biosafety, and superior biodegradability.

Chitosan-based coatings offer notable cost advantages over commercial preservatives due to their abundant natural sourcing with low procurement costs. This affordability, coupled with inherent biodegradability and regulatory compliance, positions them as a viable solution for organic farming systems where synthetic chemicals are restricted. While advanced modification technologies may elevate initial production expenses, the long-term benefits of enhanced preservation efficacy and sustainability-driven market demand outweigh these costs. Moreover, large-scale manufacturing could decrease unit costs without compromising performance, further narrowing the cost gap with traditional preservatives. These factors collectively underscore the dual economic and environmental competitiveness of chitosan-based coatings for commercial fruit preservation.

### 1.1. Chitosan

Chitosan, chemically known as β-(1,4)-2-amino-2-deoxy-D-glucose, is derived from chitin through hydrolysis or enzymatic hydrolysis under alkaline conditions to remove part of the acetyl group (degree of deacetylation > 55%), so chitosan is also called deacetylated chitin [57,58,59]. Figure 1 shows the chemical structure diagram of chitosan. Chitosan is an amorphous, translucent solid that is usually white or off-white in color. It is generally insoluble in water, sulfuric acid, phosphoric acid, alkaline solutions, and most organic solvents, but soluble in inorganic acids (e.g., hydrochloric acid, acetic acid) and many organic acids (e.g., benzoic acid) [60,61]. Under acidic conditions, the lone pair electrons on the nitrogen atom of the free amino group (-NH_2_) in chitosan molecules are protonated (-NH_3_^+^). This protonation disrupts intramolecular and intermolecular hydrogen bonds as well as the lattice structure, enabling the hydroxyl groups (-OH) to bind with water molecules and facilitating the dissolution of chitosan [62,63]. As the only natural alkaline polysaccharide in nature [64,65], chitosan has good biocompatibility and biological activity. In addition, chitosan also has good antimicrobial properties, and its structure contains a large amount of -NH_2_, which is the key to its antimicrobial properties. -NH_2_ is positively charged after being protonated (-NH_3_^+^), which can produce electrostatic interaction with negatively charged bacteria and inhibit their growth [66,67]. At the same time, chitosan also has antimicrobial activity against a variety of fungi and yeasts, which helps to reduce the number of molds in the process of fruit ripening and delay fruit decay in storage and preservation [68].

### 1.2. Modification of Chitosan

The coating made of raw chitosan is difficult to meet the diverse application needs of various objects and different conditions [69,70]. Fortunately, there are multiple active groups on the chitosan molecular chain, which are amino groups, and the primary and secondary hydroxyl groups at C-3 and C-6 positions [71,72], which facilitates the chemical modification of chitosan. Therefore, chitosan is often modified by chemical methods to improve its film-forming properties, mechanical strength, and barrier properties of the coating, and endow the coating with unique physicochemical properties, so as to meet the higher requirements of storage and preservation of various objects and different conditions. Chitosan modification technology mainly includes cross-linking modification [73], grafting modification [74], carboxylation modification [75], quaternary ammonium modification [76], etc. Cross-linking modification forms chemical bonds between chitosan molecular chains by adding cross-linking agents, thereby forming a stable three-dimensional network structure and improving the mechanical strength and water resistance of the coating [77]. Graft modification is to graft the active groups on the chitosan molecular chain with other polymer substances to form graft copolymers with multiple functions. The amino group at the C-2 position, the primary hydroxyl group at the C-6 position, and the secondary hydroxyl group at the C-3 position in the chitosan molecule can be used as grafting reaction points [78]. The carboxylation reaction of chitosan usually introduces acidic groups into the chitosan backbone to improve its solubility, moisturizing properties, and film-forming properties. The substitution order of carboxymethyl group is primary hydroxyl group at C-6 position, secondary hydroxyl group at C-3 position, and amino group at C-2 position [79]. The quaternary ammonium reaction of chitosan is mainly to introduce quaternary ammonium groups on its amino group, so as to improve water solubility and antimicrobial properties [80,81].

The key to the application of chitosan modification technology lies in the selection of appropriate modification methods and modifiers. Different modification methods and modifiers will have different effects on the film-forming properties and the mechanical strength of the coating. Therefore, in practical application, it is necessary to select the appropriate chitosan modification technology according to different storage needs and market demand and comprehensively consider the modification effect and cost-effectiveness. At the same time, it is also necessary to strictly control the preparation process and conditions of the modified coating to ensure the quality and stability of the coating [78].

### 1.3. Preservation Mechanism of Chitosan-Based Coating

Chitosan-based coatings exert their preservative effects through three interconnected mechanisms: physical barrier, antimicrobial activity, and physiological metabolic regulation.

#### 1.3.1. Physical Barrier

Chitosan can form a continuous, dense, and elastic semi-permeable coating on the surface of fruits [82]. The presence of this coating significantly blocks the gas exchange between the external environment and the inside of the agricultural products [83,84,85]. By reducing the infiltration of oxygen and the escape of carbon dioxide, the chitosan-based coating effectively reduces the intensity of respiration in the fruit, avoids nutrient consumption and energy waste caused by excessive respiration, and thus slows down the senescence rate of the fruit [86,87,88]. In addition, the presence of elastic semi-permeable membranes can reduce the damaging effect of external mechanical loads to a certain extent. At the same time, this barrier effect also limits the loss of water inside the fruit and maintains the water content, which is essential for maintaining the fresh taste and color of the fruit [89].

#### 1.3.2. Antibacterial and Antifungal Activity

Chitosan has good broad-spectrum antibacterial and antifungal activity and can inhibit the growth of harmful microorganisms on the surface of fruits [90,91,92]. The active functional groups such as amino groups rich in the chitosan molecular chain are the key to its antibacterial and antifungal effect [93]. These functional groups can attract each other to the negative charge on the cell membrane of microorganisms through electrostatic action, and then destroy the integrity of the cell membrane, resulting in the leakage of cell contents and the disintegration of cell structure, and finally achieve the effect of inhibiting or killing microorganisms [94]. This mechanism of direct action on the cell membrane of microorganisms makes chitosan show good inhibitory activity against a variety of common pathogenic microorganisms on the surface of fruits, such as molds, yeasts, and bacteria [95]. The inhibitory effect of chitosan coating on Gram-positive bacteria is generally better than that of Gram-negative bacteria, which may be due to the more complex structure of the outer membrane of Gram-negative bacteria [96]. What is more noteworthy is that chitosan can also stimulate the defense mechanism of fruits themselves. Under the action of chitosan, fruit tissues may induce the production of a series of antimicrobial and antifungal peptides and other defensive substances, and these endogenous antimicrobial and antifungal factors can further enhance the resistance of fruits to pathogenic microorganisms, forming a protective network from the inside out [52,97,98]. To sum up, during fruit storage, the chitosan-based coating significantly reduces the microbial load on the fruit surface through its direct antibacterial and antifungal effect, indirect effect of inducing the production of defensive substances in the fruit, effectively delays the decay and deterioration process caused by microbial activities, and ensures that the fruit remains fresh and safe for a longer time.

#### 1.3.3. Physiological Metabolic Regulation

The contribution of chitosan-based coating to fruit preservation is also reflected in the ability to fundamentally delay fruit ripening and senescence by affecting the physiological and metabolic processes inside the fruit. First of all, chitosan-based coating can affect the enzyme activity associated with senescence in fruits [99]. During ripening, the activity of cell wall degrading enzymes (e.g., polygalacturonidase) produced by the fruit itself gradually increases. This enzymatic upregulation directly weakens cell structural integrity and results in fruit softening. Chitosan-based coatings may directly or indirectly inhibit the activity of these enzymes and slow down the destruction of cell wall structures, thereby maintaining fruit firmness and integrity. Secondly, chitosan may also affect other enzyme activities related to fruit quality, such as the antioxidant enzyme system, which can enhance the antioxidant capacity of fruits, reduce the accumulation of free radicals, and further delay fruit senescence [100,101]. Hong et al. [102] investigated the effects of chitosan coating on the physicochemical properties of guava fruit and confirmed that the coating could induce significant increases in peroxidase, superoxide dismutase, and catalase activities, and inhibit the production of superoxide free radicals (reduced by 13%). Last but not least, chitosan-based coating also plays an important role in regulating the hormone balance of fruits. As an important hormone in plants, ethylene has a significant role in promoting fruit ripening and senescence. Chitosan-based coatings may slow down the ripening of the fruit and preserve its quality and flavor by influencing the activity of key enzymes in the ethylene synthesis pathway, or acting as a regulator of ethylene receptors, reducing the amount of ethylene produced or weakening its stimulating effect on the fruit [103,104].

It is worth noting that the regulatory effect of chitosan-based coating on fruit physiological metabolism does not exist in isolation but complements with film-forming and constitutes a complex preservation mechanism. Through the dual effects of physical barrier and biological regulation, the chitosan-based coating not only slows down the damage of the external environment to the fruit, but also optimizes the metabolic environment inside the fruit, and realizes the all-round preservation effect from the outside to the inside [105,106]. Figure 2 illustrates the preservation mechanism of chitosan-based coatings.

## 2. Application of Chitosan Coating in Fruit Preservation

### 2.1. Raw Chitosan

Due to the above advantages of chitosan, raw chitosan can be directly used in fruit coating after dissolving in acidic solution. One study compared the effectiveness of postharvest chitosan treatment (1.4%) and acetic acid treatment (1%) on Somerset Seedless grapes. The results showed that chitosan performed better in Somerset Seedless postharvest storage, and the damage rate was lower. Although acetic acid had a similar positive effect on mold control as chitosan treatment, it resulted in a higher split rate after 1 week of storage and a higher shatter rate after 5 weeks. In addition, chitosan-treated Somerset Seedless still met USDA table grape standards after 5 weeks [107]. Liu et al. [108] studied the effects of 2% chitosan coating on the preservation effects of fertile orange at 25 °C. The results indicated that chitosan significantly inhibited the decrease in weight loss, the ratio of soluble solids to acidity, soluble solids and total titratable acids, and delayed the decrease in total phenols, flavonoids, carotenoids, and ascorbic acid.

For chitosan, its deacetylation degree and molecular weight have a significant effect on the postharvest preservation effect of fruits. Zheng et al. [109] investigated the effects of chitosan coatings with different degrees of deacetylation (88.1% and 95.2%) on the postharvest characteristics and internal metabolism of sweet cherries. The results revealed that the chitosan coating effectively inhibited the changes in weight, color and hardness, maintained the total phenols contents, flavonoids and titratable acids contents, and reduced the activities of β-galactosidase and polyphenol oxidase. In addition, the preservation effects on sweet cherry were remarkably related to the contents of sorbitol, 4-hydroxycinnamic acid, tyrosine, etc. The chitosan coating with 88.1% deacetylation degrees content had better wettability and stronger metabolic regulation ability. Yuan et al. [110] evaluated the effects of low, medium, and high molecular weight (30, 150, 300 kD) chitosan coatings on the storage quality and fungal inhibition of mini cucumber. The results showed that the chitosan coating significantly inhibited the change in weight loss, color, DPPH free radical scavenging rate and total phenol content, and decreased the total number of bacterial colonies and the number of viable bacteria of yeast and mold. Specifically, chitosan coatings with low molecular weight exhibited the best preservation effect and antifungal activity and extended the shelf life of the mini cucumber by 6 days.

Furthermore, the coating process, such as chitosan concentration and film-forming time, also has a great influence on the final preservation effect. Alam et al. [111] evaluated the effect of chitosan (1%, 2%, and 3%) on the physicochemical properties of persimmon during storage at specific dipping times (5, 10, and 15 min). The results showed that maximum firmness, ascorbic acid, and titratable acidity and minimum TSS, pH, TSS/acid ratio, decay incidence, and weight loss were found in the persimmon fruits coated with 3% chitosan. Similarly, fruits that were dipped for 15 min performed best. That is, after dipping persimmon fruit in 3% chitosan solution for 15 min, storage at room temperature could improve its postharvest performance. Kaur and Nikhanj [112] adopt the response surface method to optimize the chitosan coating process. It was found that cucumbers after dipping in a 1% *w*/*v* chitosan solution for 5 min had a minimal microbial load, in addition to maintaining their physicochemical properties after 12 days, thus extending their shelf life. Moreover, chitosan-coated cucumbers can still meet the consumption standard for up to 12 days while maintaining their physicochemical, microbiological, and organoleptic properties.

However, with the increase in storage time and the change in external storage environment, raw chitosan coating is often difficult to meet the requirements of postharvest preservation of various fruits. Chemical modification of chitosan is one of the solutions, and the addition of functional groups can give it unique physicochemical properties.

### 2.2. Chemically Modified Chitosan

Chemical modification is an effective way to improve the physicochemical properties of chitosan including solubility, thermal stability, rheology, oxidation resistance, antimicrobial properties, etc. Common chemical modification methods include carboxymethylation, quaternization, grafting, and cross-linking. The synthesis of chemically modified chitosan is illustrated in Figure 3.

#### 2.2.1. Carboxymethyl Chitosan

Chitosan is poorly water-soluble and often needs to be soluble in acidic solutions, which limits its direct use in food packaging coatings. The introduction of carboxymethyl into the chitosan structure and the preparation of carboxymethyl chitosan can break the limitation of poor water solubility and expand the application scenarios. In addition, carboxymethyl chitosan has excellent biodegradability, biocompatibility, antioxidant properties, and antibacterial and antifungal activities [113]. The carboxylation of the amino and hydroxyl groups of chitosan forms *N*-carboxymethyl chitosan and O-carboxymethyl chitosan, respectively. The simultaneous carboxylation of both groups can lead to the production of N,O-carboxymethyl chitosan. The effect of chitosan particle sizes (75, 125, 250, 450, and 850 μm) on the physicochemical properties of carboxymethyl chitosan was investigated. The results showed that with the decrease in chitosan particle size, the degree of substitution and water solubility of carboxymethyl chitosan increased. The increase in water solubility was due to the higher conversion and the increased polarity of chitosan to carboxymethyl chitosan. In addition, as the chitosan particle size decreased, the tensile strength, elongation at break, and water vapor transmission rate of the carboxymethyl chitosan coating increased [114].

#### 2.2.2. Quaternized Chitosan

The quaternary ammonium modification of chitosan is mainly to introduce a quaternary ammonium group on its amino group, so as to achieve the goal of increasing the water solubility of chitosan [115]. Peng et al. [116] synthesized water-soluble quaternized chitosan by reacting chitin with (3-chloro-2-hydroxypropyl) trimethylammonium chloride in an aqueous solution of NaOH/urea. By varying the molar ratio of the substrate concentration as well as the reaction time, a water-soluble quaternized chitosan with a degree of substitution (DS) value of 0.27–0.54 can be obtained. Compared to polyethylenimine (25 kDa), quaternized chitosan exhibited excellent low cytotoxicity.

#### 2.2.3. Grafted Chitosan

To improve the antibacterial, antioxidant, and anti-fogging properties, coumaric acid grafted chitosan was synthesized by carbodiimide coupling reaction. The resulting new coating exhibited excellent transparency, UV blocking, and anti-fogging properties. Compared to chitosan coatings, coumaric acid grafted chitosan coatings exhibited stronger antioxidant capacity and antimicrobial properties against E. coli, Staphylococcus aureus, and Botrytis cinerea, and reduced the decay of strawberries at room temperature [117]. To address the problem that the antimicrobial properties of chitosan often diminish with increasing solubility, polyphenol gallic acid was conjugated to chitosan using 1-ethyl-3-(3-dimethylaminopropyl)carbodiimide/*N*-hydroxysuccinimide to prepare grafted chitosan coatings. The resulting coating exhibited excellent solubility, mechanical strength, UV blocking properties, and superior oxidation and antimicrobial properties. In addition, the coating showed a high affinity for hydrophobic fruit surfaces, while also contributing to easy cleaning [118].

#### 2.2.4. Cross-Linked Chitosan

The free -NH_2_ and -OH groups on chitosan can form amide or imine bonds with other polymers or crosslinkers, thus forming a stable three-dimensional network structure with better mechanical and barrier properties [119]. Du et al. [120] developed a cross-linked chitosan coating based on sodium tripolyphosphate as a crosslinker. The results showed that the mechanical strength, oxygen barrier properties, thermal stability, antimicrobial activity, and hydrophobicity of the coating were significantly enhanced, and it was effective in maintaining the visual appeal of bananas and prolonging their storage time.

#### 2.2.5. Multiply Modified Chitosan

In addition to the above-mentioned chemical modifications, there are esterification, hydroxyl alkylation, sulfonation, oxidation, etc. To give full play to the advantages of chitosan in the field of fruit preservation, different chemical modification methods can also be combined. For example, although quaternary ammonium modification can effectively enhance the solubility of chitosan and enhance its antimicrobial properties, the lack of adequate mechanical, oxidant, and UV resistance still severely limits its preservation performance for fruit preservation. He et al. [121] successfully prepared chitosan-based food preservation coatings with satisfied antibacterial, antioxidant, and anti-ultraviolet properties by mixing quaternary ammonium salt and tannic acid-modified chitosan and oxidized chitosan. Quaternary ammonium salt modification effectively improved the water solubility of chitosan and enhanced its antimicrobial properties. The introduction of tannic acid provided chitosan with good antioxidant and UV properties. Notably, the Schiff bond formed between different modified chitosan can effectively enhance the mechanical properties and water/oxygen barrier properties of the coating. In addition, the preservation performance, biocompatibility, and safety of the coating were verified on strawberries, bananas, and mushrooms. The grafting of *N*-carboxymethyl chitosan with gallic acid using the biological enzymatic method was an efficient and environmentally friendly molecular synthesis technique. The coating enhanced the antioxidant properties and nutrients and extended the shelf life of strawberries. Specifically, it reduced the weight loss of strawberries from 12.7% to 8.4% and the decay rate from 36.7% to 8.9% and maintained the titratable acidity content (above 60%) [122].

## 3. Application of Chitosan-Based Composite Coating in Fruit Preservation

Despite its advantages, stand-alone chitosan coatings have limitations, including poor stability in acidic environments, low mechanical strength, and insufficient water vapor barrier properties [123]. Therefore, most of the current research adopts the method of composite coating with other substances to enhance its performance and thus prepare chitosan-based composite coating. The research progress on the effect of chitosan-based composite coating in fruit preservation is shown in Table 1.

Chitosan has antimicrobial properties and good film-forming capability and can form a dense coating on the surface of the fruit to reduce the impact of the external adverse environment. However, the antimicrobial ability of chitosan itself is limited, and it is necessary to enhance the antimicrobial ability of chitosan with the help of active substances [134,135]. Bansal et al. [124] found that coatings containing 2.0% curry leaf essential oil exhibited the highest antimicrobial activity and antioxidant properties (up to 71.80%), and the lowest water solubility. The preservation performance of the coating was evaluated by storing grapes at room temperature and refrigerated conditions. It was confirmed that the coating extended the shelf life to 12 days at room temperature and to 20 days under refrigeration. Venkatachalam et al. [125] incorporated cinnamon oil into chitosan-based edible coatings, which significantly improved their antimicrobial properties. The higher concentration (>1%) of the cinnamon oil effectively modulated physicochemical qualities such as pH, TSS, TA, and TSS/TA ratios during 14-day storage, while also retaining essential phytochemicals, including chlorophyll and lycopene.

Blending with other macromolecular materials can effectively improve the mechanical and barrier properties of chitosan-based coatings. Yu et al. [126] significantly enhanced the surface properties of the chitosan coating and increased its tensile strength and elongation at break by incorporation of curdlan. In addition, the coating exhibited enhanced permeability to water vapor, oxygen, and carbon dioxide, as well as improved light transmittance. A 10-day storage experiment showed that the composite coating was significantly superior to the single coating and uncoated samples in maintaining the quality of the cherry tomatoes. Suresh et al. [127] found that the composite coating composed of almond gum and chitosan exhibited low moisture content (34.8  ±  2.02%) and solubility (42.56  ±  2.3%), and high elongation (63.2  ±  4.12%). Furthermore, coated tomatoes and blueberries showed a substantial improvement in weight loss, reduced acidity, and lower ascorbic acid levels after a shelf life of 25 and 20 days, respectively, compared to uncoated tomatoes and blueberries. Yan et al. [128] found that the incorporation of trans-cinnamaldehyde enhanced the tensile strength, light transmittance, water vapor permeability, and antimicrobial properties of the composite coating. In addition, the results showed that the composite coating delayed the respiration, weight loss, and loss of sugar content, and maintained the hardness, color, total soluble solids, titratable acid, and appearance of the banana.

## 4. Application of Chitosan-Based Nanocomposite Coating in Fruit Preservation

With the gradual development of nanoscience, nanomaterials have been widely considered to be added to chitosan-based coatings to endow the coating with antibacterial, antioxidant, enhanced mechanical strength and barrier properties, further improve the anti-corrosion, preservation and anti-damage ability of fruit, and prolong the shelf life [136,137].

The addition of nanoparticles has the following effects: (1) Destroying the microbial structure, inhibiting enzyme activity, and delaying spoilage. (2) Scavenging free radicals, chelating metal ions, and inhibiting oxidative browning. (3) Enhancing the mechanical properties of the coating and reducing the damage of storage and transportation. (4) Forming tortuous paths inside the coating to reduce the mobility of gas and water and maintaining the quality of fruits. (5) Realizing the intelligent controlled release of active substances and synergistically improving the preservation effect. For example, Zhou et al. [138] successfully synthesized resveratrol-encapsulated shellac nanoparticles by antisolvent co-precipitation. The mechanical properties of the prepared coatings were significantly improved, and the coated resveratrol enhanced the antimicrobial activity. Moreover, this coating extended the shelf life of strawberries to 15 days at 4 °C. The research progress on the effect of chitosan-based nanocomposite coating in fruit preservation is shown in Table 2.

Inorganic nanoparticles come from a wide range of sources and are often used as reinforcing agents in polymeric materials. Yin et al. [139] successfully prepared κ-carrageenan/carboxymethyl chitosan/arbutin/kaolin clay nanocomposite coatings. The incorporation of kaolin clay into the coating reduced the permeability of water vapor and oxygen and enhanced the water resistance. In addition, the tensile strength of the nanocomposite coating significantly improved. With the addition of kaolin clay, the tensile strength increased from 20.60 MPa to 34.71 MPa. It was also demonstrated that the nanocomposite coating could delay fruit ripening, respiration, dehydration, and microbial invasion, thus extending the storage of cherry tomatoes at 28 °C for 9 days. Tayel et al. [140] constructed a nanocomposite coating of chitosan/fenugreek seed mucilage/selenium nanoparticles, which exhibited excellent antifungal effects against P.digitatum isolates, with an inhibition diameter of 32.2 mm and an inhibitory concentration of 12.5 mg/mL. In addition, the infected lemons completely eliminated the green mold on the surface after 10 days of coating. Taghipour et al. [141] coated fresh pistachios with chitosan/TiO_2_ nanocomposite coating and stored them at 4 ± 0.5 °C. It was found that the shelf life of fresh pistachios can be extended by up to 30 days, effectively preserving the nutrient content, organoleptic quality, nutritional value, and antioxidant capacity of fresh pistachios. When ZnO was used as a nanofiller instead of TiO_2_, the shelf life of fresh pistachios can be increased to 35–40 days [142].

In addition to inorganic nanoparticles, the reinforcement effect of biomacromolecule-derived organic nanoparticles on chitosan coatings has also received extensive attention, especially considering that the coatings are in direct contact with food, and the biosafety of nanomaterials still needs to be carefully evaluated. Wardana et al. [146] found that the addition of chitosan nanoparticles significantly enhanced antifungal efficacy, with a 65% reduction in Botrytis cinerea activity compared to pure chitosan. The incorporation of chitosan nanoparticles improved ultraviolet and visible wavelengths, water vapor permeability, hydrophobicity, and thermal properties. In addition, the coating reduced the tendency of color and TSS. Yuan et al. [147] prepared a double-layer nanocomposite coating based on a layer-by-layer self-assembly method. The outer layer of quaternized chitosan was positively charged, which significantly enhanced the antimicrobial properties. The inner layer of aldehyde carboxycellulose nanofibers had excellent mechanical properties and adhered tightly to the peel, preventing cracking or peeling. Quaternized chitosan and aldehyde carboxycellulose nanofibers can be cross-linked by electrostatic interactions and Schiff base reactions to form a dense interfacial layer. In addition, the chitosan-based nanocomposite coating can effectively reduce the water loss, delay softening, shrinkage and decay, and effectively prolong the shelf life of the fruits by 5–10 days. Sharma et al. [148] investigated the physicochemical properties of different nanoparticle-reinforced nanocomposite coatings, and the results showed that the performance of starch nanocrystals-reinforced coatings was better than that of montmorillonite-reinforced coatings because of the superior barrier properties of starch nanocrystals. Additionally, the coating can effectively maintain acceptable lychee quality for three weeks.

It is worth noting that although the use of chitosan-based coating helps to maintain the sensory quality of the fruit during storage, it is still necessary to pay attention to the effects of the coating on the color, smell, and taste of the fruit itself, so as not to affect the purchase preferences of consumers. For example, Han et al. [151] confirmed via sensory evaluation that the chitosan coating did not change the sweetness or flavor acceptance of strawberries but may cause a slight astringency due to excessive coating concentration (e.g., 1.5%). This highlights a concentration-dependent balance between preservation and sensory appeal, necessitating formulation optimization. In addition, environmental factors can also affect the efficiency of chitosan coatings. Hesami et al. [152] found that chitosan coatings had a greater positive effect on the shelf life and weight maintenance of ber fruit at low temperature (5 °C) than at high temperature (25 °C). This was due to the fact that the low temperature has a synergistic effect with the preservation ability of the chitosan coating by inhibiting fruit respiration and microbial metabolism. In addition, the pH of the coating solution exerts a pivotal influence on the dissolution behavior and structural integrity of chitosan. Specifically, an acidic environment (pH 4–6) optimizes chitosan functionality by promoting protonation of amine groups, which enhances solubility and strengthens electrostatic interactions with negatively charged microbial cell membranes, thereby improving antimicrobial efficacy. Conversely, at pH > 6.5, deprotonation of chitosan reduced its solubility and compromised film-forming capacity, resulting in impaired antimicrobial efficacy and reduced adherence to fruit surfaces [153].

## 5. Challenges and Perspectives

In the early days, a single chitosan coating had defects such as low mechanical strength and narrow antimicrobial spectrum, but now it has achieved a leap in performance through molecular modification, natural compounding, and nanomaterial synergistic enhancement. The application boundary of chitosan-based coating continues to expand, showing a broad prospect for industrialization.

At present, there are still great challenges in the chitosan-based coating of fruits. Primary challenges include the time-consuming and labor-intensive nature of fruit preservation assays, compounded by the absence of standardized metrics for quantifying coating efficacy. Secondly, there is a lack of efficient and fast coating equipment. For example, the dip coating method needs to control the concentration and time, and the accuracy of the spraying equipment directly affects the consistency of coating thickness. Finally, the influence of environmental factors on the efficiency of chitosan coatings still needs to be further studied. For instance, high temperature and high humidity areas have higher requirements for the moisture permeability of the coating. Therefore, in view of the above problems and challenges, future research prospects include the optimization of coating formulations based on machine learning, the development of intelligent coating equipment, and the study of coating adaptability in different environments.

## 6. Conclusions

Fruit preservation technology is undergoing a transformation from chemical methods to natural, safe, and efficient methods. As a natural polysaccharide, chitosan has become one of the research hotspots in the field of fruit coating preservation due to its film-forming ability, antibacterial properties, biocompatibility, and degradability. Chitosan coating prolongs the shelf life of fruits through the dual effects of physical barrier and physiological regulation. On the one hand, its porous network structure forms a selective permeable membrane that reduces oxygen permeability (inhibits respiration) while allowing carbon dioxide diffusion and retarding ethylene synthesis. On the other hand, the cationic properties of chitosan destroy the microbial cell membrane and combine with the adsorption and fixation of water by the coating to form an antimicrobial and moisturizing microenvironment. Furthermore, the network structure formed by chitosan serves as a carrier for active substances, synergistically enhancing antimicrobial and antioxidant capabilities.

At present, the research on chitosan fruit coating mainly focuses on functional modification and co-preservation. For example, the combination of inorganic/organic nanofillers to enhance mechanical strength and gas barriers, and the combination of plant polyphenols/essential oils to enhance antimicrobial and antioxidant properties, break through the limitations of a single chitosan coating that is brittle and has poor antimicrobial effect. However, at present, there is still a lack of efficient fruit and vegetable coating equipment, reducing the cost of industrialization, and realizing large-scale preparations is still the key to industrialization. In the future, it will be possible to make a technological leap from the laboratory to the postharvest processing line by combining genetically modified chitosan directional modification or microfluidic coating technology. Furthermore, the effect of environmental conditions on chitosan coating performance requires additional research to advance industrial scalability.

## Figures and Tables

**Figure 1 foods-14-01318-f001:**
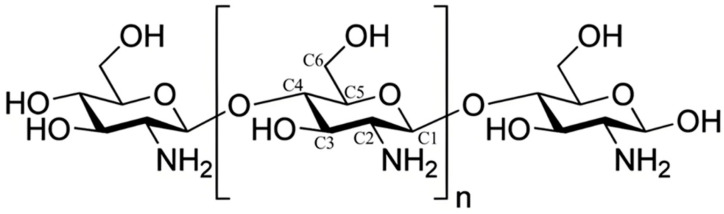
Chemical structure diagram of chitosan.

**Figure 2 foods-14-01318-f002:**
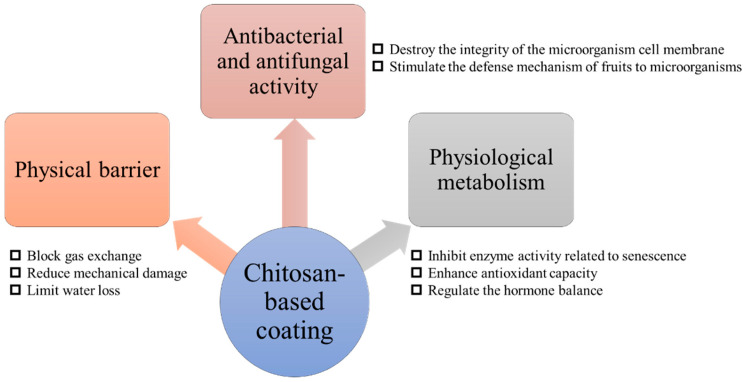
Schematic diagram of chitosan-based coating preservation mechanisms.

**Figure 3 foods-14-01318-f003:**
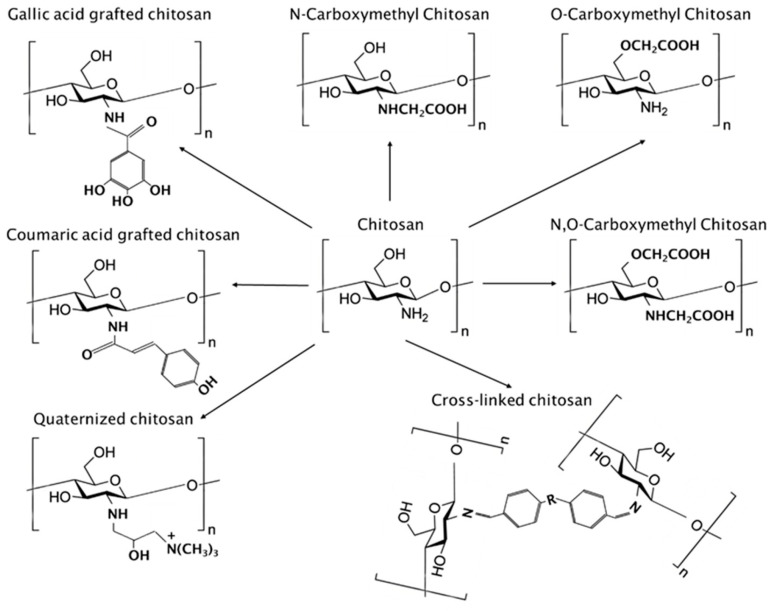
Systematic diagram of synthesis of chemically modified chitosan.

**Table 1 foods-14-01318-t001:** Advances in the effects of chitosan-based composite coating in fruit preservation.

Fruit	Composition	Effect	Reference
Green grapes	Buckwheat starch + chitosan + curry leaf essential oil	Extend the shelf life to 12 days at room temperature and to 20 days under refrigeration storage.	[124]
Tomato	Chitosan + cinnamon oil	Modulate physicochemical qualities such as pH, TSS, TA, and TSS/TA ratios during 14-day storage.	[125]
Cherry tomato	Chitosan + curdlan	Maintain the quality of the postharvest cherry tomatoes at 10 days of storage.	[126]
Tomatoes and blueberry	Almond gum + chitosan	Extend the shelf life of tomatoes and blueberries to 25 days and 20 days.	[127]
Banana	Chitosan + diepoxy-polyethylene glycol + trans-cinnamaldehyde	Maintain the levels of TSS, TA, and appearance within 12–24 days.	[128]
Guava	Chitosan + carvacrol	Maintain the quality of guava with higher hardness, SSC, TA, and total phenol content, and lower weight loss.	[129]
Cherry tomato	Carboxymethyl chitosan + polycaprolactone	Maintain the weight loss, firmness loss, and color deepening during storage.	[130]
Persimmon	Chitosan + ginger oil	Inhibit changes in weight loss, respiration rate, ethylene formation, pH, and TSS during storage.	[131]
Saimaiti apricot	Chitosan + chitosan grafted with gallic acid	Extend the shelf life by 12 days compared to traditional refrigeration.	[132]
Strawberry	Carboxymethyl chitosan + oxidized carboxymethyl cellulose	Reduce the decay rate by about 42% compared to the control.	[133]

**Table 2 foods-14-01318-t002:** Advances on the effects of chitosan-based nanocomposite coating in fruit preservation.

Fruit	Composition	Nanomaterial	Effect	Reference
Strawberry	Chitosan + carnauba wax	Resveratrol-encapsulated shellac nanoparticles	Enhance the mechanical and UV resistance properties and extend shelf life to 15 days.	[138]
Cherry tomato	Carboxymethyl chitosan + κ-carrageenan + arbutin	Kaolin clay	Enhance tensile strength from 20.60 MPa to 34.71 MPa and prolong storage for 9 days at 28 °C.	[139]
Lemon	Chitosan + fenugreek seed mucilage	Selenium nanoparticles	Eliminate green mold development after 10 days of coating.	[140]
Fresh pistachio	Chitosan	TiO_2_	Extend the shelf life to 30 days.	[141]
Fresh pistachio	Chitosan	ZnO	Extend the shelf life to 35–40 days.	[142]
Avocado	Chitosan + arabic gum	Zinc nanoparticles	Inhibit weight loss, decay, and improve peel and pulp color.	[143]
Papaya	Chitosan + cassava starch	TiO_2_	Reduce weight loss by 7.12 ± 1.57% and 5.27 ± 0.31% in light and darkness, respectively.	[144]
Passion fruit	Chitosan + beeswax	Graphene oxide	Enhance SSC and TA by 16.7% and 31.9% on day 8.	[145]
Tangerine	Chitosan	Chitosan nanoparticles	Inhibit the tendency of color and TSS.	[146]
Mango, banana and loquat	Quaternized chitosan	Aldehyde carboxycellulose nanofibres	Extend the shelf life of fruits by 5–10 days.	[147]
Lychee	Gluten + chitosan	Starch nanocrystals/montmorillonite	Maintain acceptable quality for three weeks.	[148]
Strawberry	Chitosan + dialdehyde carboxymethyl cellulose	Zein-loaded cinnamaldehyde nanoparticles	Extend the shelf life to 7 days.	[149]
Strawberry	Chitosan + raspberry leaf extract	Lignocellulosic nanofibers	Reduce weight loss by 26.4% compared to the uncoated group.	[150]

## Data Availability

No new data were created or analyzed in this study. Data sharing is not applicable to this article.

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
