# Peer review of "Application of Chitosan and Its Derivatives in Postharvest Coating Preservation of Fruits"

_foods, 2025, doi:10.3390/foods14081318_

Round 1
Reviewer 1 Report
Comments and Suggestions for Authors
Dear Authors.
The topic of manuscript is interesting in the context of identification new and sustainable packaging solutions to reduce the utilisation of plastic packaging.
The paper is well structured with suggestive abstract and conclusions. However some improvements before the publishing are necessary :
- A short description of antifungal activity of chitosan within Section 1.3., it is important in the context of its use as fresh fruits packaging.
- A section about the utilisation of chitosan in coatings for packaging paper to confer the functional properties (barrier to water, oxygen, ..) will complete whole the applications of chitosan as agro-products (including fruits) packaging. This is supported by that the coated papers or boards can be used as trays for fresh fruits.
Reviewer 2 Report
Comments and Suggestions for Authors
In this work, a review on the main properties of chitosan and its application for fruit storage is presented. The sequence is appropriate and basic properties of the biopolimer are provided. The information found here may be of the interest for chitosan researchers.
In my opinion, a carefull revision of the manuscript should be done in order to impove the fluency. Some suggestions are the following:
In the 1.1 section, improve the sentence “Under acidic conditions, the lone pair electrons of N atoms of free amino group (-NH2) in chitosan molecule is protonated (-NH3+), which destroys the intramolecular or intermolecular hydrogen bond and lattice structure, so that -OH binds to water molecules, thereby realizing the dissolution of chitosan [59, 60]”; it is complicated to read.
In the 1.3.3 section the sentence “During the ripening process of fruit, the enhancement of the activity of cell wall degrading enzymes such as polygalacturonidase is the main reason for the softening and decreasing firmness.” Is confused.
Reviewer 3 Report
Comments and Suggestions for Authors
Dear authors, below are some considerations with my suggestions for improving the work.
Please, reduce the amount of wording duplication in the manuscript.
Abstract: could better emphasize the novelty of this review compared to previous works. In addition, please include a brief mention of the environmental impact and industrial scalability
Keywords: please improve this section and avoid using words that were already utilized in the title
Please include some comparison with other preservation techniques beyond a brief mention in the introduction.
An economic feasibility discussion is missing, what are the cost implications of using chitosan-based coatings compared to other commercial preservatives? The paper doesn`t sufficiently discuss how scalable these applications are for commercial fruit preservation.
What is the consumer perception and regulatory constraints regarding chitosan applications in food?
There are environmental factors (e.g., humidity, temperature variations) that affect the efficiency of chitosan coatings?
Some studies lack statistical context, effect sizes should be mentioned where possible?
The description of antimicrobial action is too general; which microbial species are most affected by chitosan?
The review presents useful interpretations but does not critically analyse contradictory findings in the literature.
Are some fruit varieties more compatible with chitosan?
How about the effects of chitosan coatings on fruit flavour and texture?
There are interactions between chitosan and other bio-based coatings?
There is a limited discussion on environmental impact, particularly in relation to biodegradability and potential ecological consequences.
Introduction: please add a paragraph discussing the environmental and economic challenges of current preservation methods
Conclusion: I suggest to include some information in the areas of environmental impact and industrial scalability.
Reviewer 4 Report
Comments and Suggestions for Authors
General comments
This review studies the application of chitosan and its derivatives in postharvest coating preservation of fruits. The study can be of interest to the field of fruit preservation. It is well written, covers a wide range of aspects and involves an extense and recent bibliography. the food industry. A few changes and clarifications and changes can improve the quality of this review. Specific comments were given hereinafter.
Specific comments
Introduction
At lines 34-36, the authors mentioned that physical preservation technology mainly includes low temperature, modified atmosphere, sonication, and irradiation treatment methods, which has the advantages of simple operation. However irradiation did not require simple operation. I suggest the authors to revise this paragraph, also at lines 35 and 42, the use of “this method” is not clear.
At lines 111-114, it is mentioned that chitosan-based coatings exert their preservative effects through three interconnected mechanisms: physical barrier, antibacterial activity and physiological metabolic regulation. But it is omitted the antioxidant activity and the same happens in Figure 2. I suggest to add info about this issue.
Application of chitosan coating in fruit preservation
At line 179, It is not clear which are the two preservation methods, clarify this point.
Regarding the application to fruits, the effect of chitosan coatings on sensory acceptability is not consider and this point is a key factor for practical use of chitosan coatings. I suggest to discuss this aspect.
Round 2
Reviewer 3 Report
Comments and Suggestions for Authors
The authors responded to my suggestions and they have addressed all the comments appropriately. In my opinion, the manuscript is now ready for acceptance.